

# First observations of ovary regeneration in an amphipod, *Ampelisca eschrichtii* Krøyer, 1842

Valentina B. Durkina[1], John W. Chapman[2] and Natalia L. Demchenko[3]

[1] Laboratory of Physiology, A.V. Zhirmunsky National Scientific Center of Marine Biology, Far Eastern Branch, Russian Academy of Sciences, Vladivostok, Primorsky Krai, Russia
[2] Department of Fisheries, Wildlife and Conservation, Oregon State University, Newport, Oregon, United States of America
[3] Laboratory of Marine Ecosystem Dynamics, A.V. Zhirmunsky National Scientific Center of Marine Biology, Far Eastern Branch, Russian Academy of Sciences, Vladivostok, Primorsky Krai, Russia

Corresponding author
John W. Chapman,
john.chapman@oregonstate.edu

## ABSTRACT

**Background:** Females of the gammaridean amphipod *Ampelisca eschrichtii* with signs of regenerating, previously atrophied ovaries were recovered from the northeastern shelf of Sakhalin Island (Okhotsk Sea, Russia). Ovarian regeneration was previously unknown for any amphipod species. *A. eschrichtii* have a predominantly 2-year life cycle (from embryo to adult death) and reproduce once between late winter or early spring at the age of 2 years. Occasionally, females survive to a third year. An adaptive value of extended survival among these females is likely to require that they are also reproductive.

**Methods:** Histological sections from a second-year female with ovarian atrophy, a female with normal ovaries, a third-year female with ovarian regeneration, as well as testes of an immature and a sexually mature male were compared to determine the sources of cells of the germinal and somatic lines necessary for ovarian regeneration.

**Results:** Ovarian regeneration in the third-year female began with the formation of a new germinal zone from germ cells preserved in the atrophied ovaries and eosinophilic cells of the previously starving second-year female. Eosinophilic cells form the mesodermal component of the germinal zone. A mass of these cells appeared in the second-year female that had atrophied ovaries and in large numbers on the intestine wall of the third-year female with regenerating ovaries. These eosinophilic cells appear to migrate into the regenerating ovaries.

**Conclusions:** All germ cells of the second-year female are not lost during ovarian atrophy and can be involved in subsequent ovarian regeneration. Eosinophilic cells involved in ovarian regeneration are of mesodermal origin. The eosinophilic cell morphologies are similar to those of quiescence cells (cells in a reversible state that do not divide but retain the ability to re-enter cell division and participate in regeneration). These histological data thus indicate that eosinophilic and germ cells of third-year females can participate in the regeneration of the ovaries to reproduce a second brood. The precursors of these third-year females (a small number the second-year females with an asynchronous [summer] breeding period and ovaries that have atrophied due to seasonal starvation) appear to possess sources of somatic

and germ cells that are sufficient for ovarian regeneration and that may be adaptations to starvation stress.

## INTRODUCTION

Comparisons of length frequency distributions and reproductive development of summer and early fall populations *Ampelisca eschrichtii* Krøyer, 1842 sampled from the "Offshore" western gray whale feeding area of the northeastern Sakhalin Island Shelf over sequential years between 2002 and 2013 revealed that growth and reproduction of these populations is minor in summer relative to overall annual growth and reproduction (*Demchenko et al., 2016*). By default, growth and reproduction must therefore be predominantly in winter, when these populations have never been sampled (*Demchenko et al., 2016*). *Demchenko et al. (2016)* additionally resolved that *A. eschrichtii* are gonochoristic and have a predominantly 2 year life history that ends with female reproduction in their second year of life. Although not observed directly, most the second-year *A. eschrichtii* females thus appear to mate in late fall or winter and die after producing a single brood before summer. A few females collected in 2013 however, had clearly survived to 3 years and showed signs of regeneration of previously atrophied ovaries (*Durkina, Chapman & Demchenko (2018)*). Reduced summer growth and reproduction is consistent with summer food limitation but these populations have not been accessible for sampling in winter for more direct tests of *Demchenko et al.'s (2016)* default conclusion, that they grow and reproduce predominantly in winter. *Durkina, Chapman & Demchenko (2018)* provided additional histological evidence of trophic stress that partially corroborate summer starvation. We use additional histological data below to test whether *A. eschrichtii* are capable of ovary regeneration as a possible adaptation to seasonal starvation and, thus additionally test the *Demchenko et al. (2016)* and *Durkina, Chapman & Demchenko (2018)* default conclusions of summer starvation.

Amphipods extrude their ova into an external brood pouch where they are fertilized to form embryos that are brooded until they hatch. Different stages of development of amphipod reproductive cells are readily distinguished by histological methods (*Charniaux-Cotton & Payen, 1985*; *Demchenko et al., 2016*; *Durkina, Chapman & Demchenko, 2018*). *Durkina, Chapman & Demchenko (2018)* histological examinations of these cells revealed high prevalences of undersized and damaged oocytes, undersized broods, a lack of females brooding fully formed juveniles, atrophied ovaries, and males with mature sperm but lacking fully developed secondary sex morphologies associated with their pelagic mating. The resorption of the second generation of vitellogenic ova (atresia) in the ovaries of females that are brooding embryos in the summer (*Durkina, Chapman & Demchenko, 2018*) is consistent with summer starvation. Starvation causes destruction of immature vitellogenic oocytes in the barnacle *Balanus amphitrite* (*Sastry,*
*1983*). The predatory copepod *Paraeuchaeta norvegica* females resorb oocytes and discontinue the breeding cycle when feeding conditions are inadequate (*Sastry, 1983*). Species, with breeding periods that coincide with the available food, successfully combine embryo bearing and the production of the next batch of eggs. This is noted in *Orchestia gammarella* (*Charniaux-Cotton & Payen, 1985*) and in omnivorous *Gammarus pulex* and *G. fossarum* which have reproductive periods that extend throughout the year (*Chaumot et al., 2015*). *Durkina, Chapman & Demchenko (2018)* and thus partially corroborate *Demchenko et al. (2016)* default assumptions of trophic stress and starvation in the summer.

Resorption of second generation vitellogenic oocytes leads to ovarian atrophy (*Durkina, Chapman & Demchenko, 2018*). Resorption of ova in summer is likely to be adaptive only if it permits increased survival of brooding females or additional reproduction when more food is available. Atrophy of ovaries is likely be adaptive only if they can be regenerated and if the females surviving to a third year can produce a second brood.

Our histological examinations of the third-year *A. eschrichtii* female herein test for ovary regeneration and thus *Demchenko et al. (2016)* hypothesis of summer food limitation and resorption of ova as a survival adaptation proposed by *Durkina, Chapman & Demchenko (2018)*. Our examinations also test whether ovarian regeneration, a previously unknown reproductive adaptation in gammaridean amphipods, could occur and thus could be an adaptation to variable food sources. Our investigation of the ovarian regeneration process in *A. eschrichtii* below also expands our understanding of the regenerative potentials of amphipod crustaceans.

The *A. eschrichtii* ovaries, as in all studied amphipods (*Charniaux-Cotton & Payen, 1985*), extend dorsolaterally to the digestive tract from the second to the seventh thoracic segment. Vitellogenic oocytes form over the length of the ovaries and are the dominant element of the ovaries of mature summer and autumn females. Previtellogenic oocytes are located at the base of vitellogenic oocytes. The previtellogenic and vitellogenic oocytes are covered with follicular epithelium (derived from mesoderm cells) and form, respectively, primary and secondary follicles. Proliferation of oogonia is a rapid, difficult to observe, process (*Subramoniam, 2016*) that occurs only among mesodermal cells in a germinal zone (*Charniaux-Cotton & Payen, 1985*). The oogonia leave the germinal zone after a series of mitotic divisions and then enter a meiosis prophase to become previtellogenic oocytes. The germinal zone and the follicular epithelium are preserved in the ovaries throughout the life of amphipods (*Charniaux-Cotton & Payen, 1985*), but *Durkina, Chapman & Demchenko (2018)* had not yet found a clear germinal zone in reproductive *A. eschrichtii* ovaries.

Repair of damaged organs and tissues in animals often occurs due to stem cells (*Stoltz et al., 2015*; *Mahla, 2016*) which are undifferentiated, can self-renew and can produce differentiated descendants. Germline stem cells (GSCs) are the source of gametes in invertebrates (fruit fly) and vertebrates (medaka, zebrafish, prosimian primates, mice) (*Lin, 1998*; *Dansereau & Lasko, 2008*; *Dunlop, Telfer & Anderson, 2014*; *Grieve et al., 2015*; *Truman, Tilly & Woods, 2017*). In the ovary and testis of *Drosophila*, for example, each GSC division produces a daughter GSC, and a differentiated daughter cell (*Lin, 1998*).

Stem cells remain in microenvironments or niches—adjacent to specialized somatic cells whose signals regulate stem cell function (*Spradling, Drummond-Barbosa & Kai, 2001*). Stem cells also occur on basement membranes, and function in response to extracellular matrix signals (*Xie & Li, 2007*). *Gilboa & Lehmann (2004)* observed that GSCs in *Drosophila* can be derived from primordial germ cells (PGCs), which populate the anlagen of the gonad during embryonic development. PGC origins in amphipods were resolved in *Orchestia cavimana* (*Wolff & Scholtz, 2002*) and *Parhyale hawaiensis* (*Extavour, 2005*). PGCs come from a single cell, which is the smallest at the 8th cell stage of embryo development (g-blastomere) (*Gerberding, Browne & Patel, 2002*). The fates progenitors of PGCs are determined by the localization of germline determinants that they contain (*Extavour, 2005*). Removal of the g-blastomere in *P. hawaiensis* embryos impedes the formation of PGCs. Adults obtained from these embryos however, are fertile and produce offspring (*Modrell, 2007*). *Modrell's (2007)* results suggest an empty GSC niche remains in the somatic tissues of g-ablated juvenile *P. hawaiensis* ovaries. A signal from an empty GSC niche in the amphipods appears to recruit surrounding somatic gonad cells to become GSCs (*Kaczmarczyk, 2014*). The germline-replacing cells in g-ablated *P. hawaiensis* were of mesodermal origin (*Winchell et al., 2017*).

The mesoderm, the middle germ layer produced during embryonic development (*Kimelman & Griffin, 1998*), is the source of many mature crustacean somatic tissues including the gonads, muscles, connective tissue, the vascular system and parts of the excretory organs (*Saxena, 2005*). The undifferentiated genital apparatus of amphipods is composed of two thin strands of mesodermal cells in the postembryonic period (*Charniaux-Cotton & Payen, 1985*), while the mesodermal component of 18 mm male *A. eschrichtii* testes are readily apparent and resemble wide cords (*Durkina, Chapman & Demchenko, 2018*). The transformation of thin mesodermal strands into wide cords during the growth of male amphipods can result from mesodermal cell multiplication in the testes themselves, or from migration of mesodermal cells into the testes from the outside. These same mechanisms are likely to occur in the formation of the mesodermal component during ovarian regeneration.

## MATERIALS AND METHODS

We examined a 32 mm length female with apparent ovarian regeneration, a 24 mm length female with ovarian atrophy, a 24 mm length female with normal ovaries, two, 13 and 18 mm length females with immature ovaries, an immature 13.5 mm length male and a sexually mature 18 mm length male for this investigation. Our histological preparations were the same as (*Durkina, Chapman & Demchenko, 2018*): dewaxed 10 μm thick sections stained with hematoxylin and eosin followed by dehydration, clearing and cover-slipping. Hematoxylin-eosin staining reveals the cytoplasm of cells and their nuclei and nucleoli. Cytoplasm absorbs eosin and turns pink. Nuclei and nucleoli absorb hematoxylin and turn purple. The somatic component of the germinal zone of the ovaries and testes of *A. eschrichtii* (mesoderm cells) absorbed only eosin. Outside the gonads, we also found cells with a pink cytoplasm and a large optically empty nucleus (with filamentous chromosomes), that we refer to from here on as eosinophilic cells. We examined immature

female *A. eschrichtii* ovaries to identify their germinal zone and to clarify whether this structure is preserved during ovarian atrophy in starving females. The testes, in contrast to the ovaries, contain many more mesoderm cells (*Charniaux-Cotton & Payen, 1985*) which make the formation of the somatic component easier to observe than in the ovaries. We observed germinal zone formation in the testes of immature males (rather than in the ovaries of immature females) to understand how the somatic (mesodermal) component of the germinal zone is formed during ovarian regeneration.

*Demchenko et al. (2016)* third-year >30 mm length females were collected only in summer 2013 in samples from Site B61_13. Among the 2013 Site B61_13 females, we recognized apparent ovarian regeneration in two of the 32 mm third-year specimens and ovarian atrophy in two (24 and 27 mm length) second-year *A. eschrichtii* females carrying embryos (*Durkina, Chapman & Demchenko, 2018*, Table S1). We therefore based these analyses on cross comparisons of immature male and female *A. eschrichtii* with the third year female, to establish how ovarian regeneration occurs, as well as to identify the origins and role of eosinophilic cells in the process. We measured germ and eosinophilic cell positions and estimated cell volumes *V* (Data S1) from photographs of the histological sections using Videotest (http://www.videotest.ru; VideoTesT Ltd., St. Petersburg, Russia). We estimated ova volumes by the relation:

$$V = \frac{4}{3}\pi R r^2 \tag{1}$$

where: *R* is the radius of the long axis and *r* is the radius of the short axis of the cell. The one-way ANOVA (Data S1, Table S1) was performed in statistical programme PAST (*Hammer, Harper & Ryan, 2001*).

## RESULTS

### The germinal zone in immature and atrophied ovaries

The germinal zone of immature 13–18 mm females ovaries is well defined (Figs. 1A and 1B) and the mesoderm cells of the germinal zone are readily stained with eosin. Mesoderm cells have a rounded shape, a thin rim of cytoplasm, a large optically empty nucleus with thin filaments of chromatin and they lack a nucleolus (Fig. 1A). Follicular epithelial cells are derivatives of mesoderm and, unlike mesodermal cells, have nuclei that are intensely stained with hematoxylin. Germ cells are commonly in pairs, next to the mesodermal cells (Fig. 1B). Oogonial mitoses were also occasionally apparent in the germinal zone (Fig. 1B). A pair of germ cells (Fig. 1C) and a pair of oogonia that formed after mitosis (Fig. 1D) and previtellogenic oocytes, were apparent in the atrophied 24 mm female ovaries (Figs. 1C and 1D). The typically rounded mesodermal cells (Fig. 1A) appear deflated in atrophied ovaries (Fig. 1D). The volumes of the few germ cells in female with ovarian atrophy were slightly less but not significantly different than the smallest germ cells of immature females ovaries (Fig. 2, Data S1). Atrophied ovaries thus appear to retain elements of the germinal zone—germ cells and mesoderm cells.

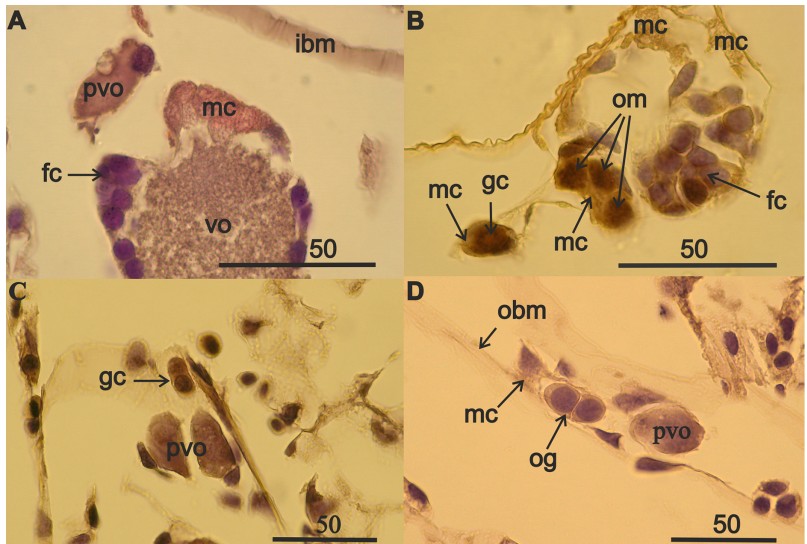

**Figure 1 Components of the germinal zone of *A. eschrichtii* ovaries.** (A) Mesodermal cells (mc), in contrast to basophilically stained follicular cells (fc), are intensely stained with eosin in the ovary of an immature 13 mm female. The ovary contains previtellogenic (pvo) and vitellogenic (vo) oocytes. (B) A germ cell pair (gc) (one out of focus), mesodermal cells (mc) and oogonial mitoses (om) in ovary of an immature 18 mm female. (C) Germ cells (gc) and previtellogenic oocyte (pvo) and (D) mesodermal cell (mc), oogonia (og) and previtellogenic oocyte (pvo) on the basal membrane of atrophied ovary of a second year 24 mm female. Scales are in μm.

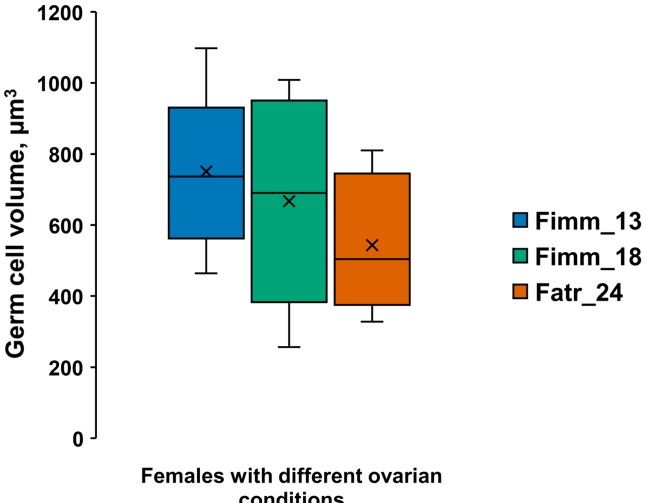

**Figure 2 Germ cell volumes (μm$^3$) [range, upper and lower quartile (box), mean (x), median (solid line)] in *A. eschrichtii*.** (Fimm_13) 13 mm immature female, (Fimm_18) 18 mm immature female and (Fatr_24) 24 mm female with ovarian atrophy. These cell volumes are not significantly different (ANOVA, $p = 0.28$).

## Mesodermal component formation in immature male testes and a third-year female regenerating ovaries

The mesodermal component of immature testes and of regenerating ovaries form in similar ways. The testes of the 13.5 mm male (Fig. 3A, t, double arrow) adjoined the

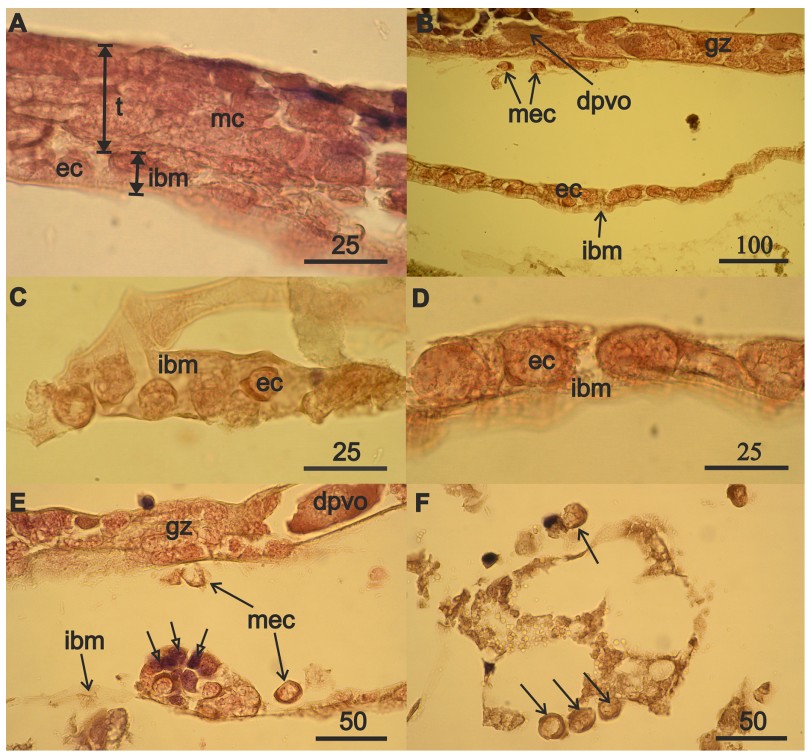

**Figure 3 Mesodermal component formation in *A. eschrichtii* gonads.** (A) Immature 13.5 mm male testis (t) containing mesodermal cells (mc) contacts the intestinal wall, which is presented by its middle layer—of basal membrane (ibm), occupied by eosinophilic cells (ec). The cells of the outer layer of the intestinal wall are rare, and its inner layer of ectoderm cells is torn off. (B) The accumulation of eosinophilic cells (ec) on the intestinal basal membrane (ibm) is located opposite the germinal zone (gz) of the regenerating ovary, putatively migrating eosiniphilic cells (mec) near the germinal zone and disintegrating deformed previtellogenic oocyte (dpvo) (upper left). (C) Eosinophilic cells (ec) on the intestinal basal membrane (ibm) of an immature 13.5 mm male. (D) Eosinophilic cells (ec) on the intestinal basal membrane (ibm) of a female with regenerating ovary. (E) Nest of eosinophilic cells on the intestinal basal membrane (ibm) of a female with regenerating ovary contains small cells formed after mitosis (open head arrows), a disintegrating previtellogenic oocyte (dpvo) (upper right) in the germinal zone, and putatively migrating eosinophilic cells (mec) (solid line arrows). (F) Putatively migrating eosinophilic cells (mec) (arrows) in adipose tissue of a female with regenerating ovary. Scales are in μm.

intestinal wall in the 2nd and 3rd thoracic body segments. The intestinal wall (from histological sections of males and females) is represented only by its middle layer, the intestinal basal membrane (ibm) (Fig. 3A, double arrows).

The nuclei in the eosinophilic mesodermal cells in the testes almost fill their cell volumes (Fig. 3A). The nucleus in eosinophilic mesodermal cells does not contain a nucleolus and chromatin forms a network structure. The mesodermal cells also are part of the germinal zone of the regenerating ovaries (Fig. 3B) and have the same characteristics as the mesodermal cells of the immature testes. Mitoses were not found among mesoderm cells in the immature testes or in the regenerating ovaries. Degrading deformed previtellogenic oocytes from the old ovary remained in the germinal zone of the regenerating ovaries (Fig. 3B).

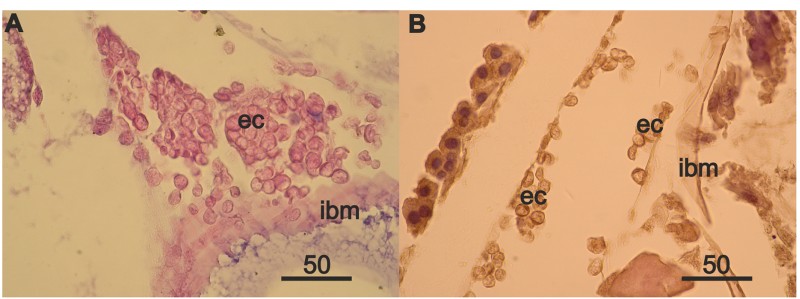

**Figure 4 Anterior part of body of 2 year _A. eschrichtii_ females with different ovarian conditions.** (A) Eosinophilic cell (ec) masses adjacent to and on the intestinal basal membrane (ibm) in female with an atrophied ovary. (B) Eosinophilic cells (ec) in comparatively sparse numbers in female with normal ovaries. Scales are in μm.                     

The formation of the mesodermal component of immature testes and the germinal zone of regenerating ovaries was accompanied by the appearance of eosinophilic cells on the intestinal basal membrane of a 13.5 mm male (Fig. 3A) and 32 mm of a 3-year-old female (Fig. 3B). Eosinophilic cells were morphologically similar to the mesodermal cells of immature testes and the germinal zone of the ovaries of the aforementioned individuals. Eosinophilic cells have a narrow rim of cytoplasm, large nuclei that lack a nucleolus, filamentous chromatin, and are small in males (Fig. 3C) compared to the females (Fig. 3D). These cells often have lamellipodia (wide processes of the cytoplasm), that are associated with cell motility. Eosinophilic cells are abundant on the intestinal wall opposite the germinal zone in female with regenerating ovaries (Fig. 3B). Reproduction of eosinophilic cells appears to occur in nests (cell groups) on the intestinal basal membrane (Fig. 3E, open arrows).

The absence of mitoses among eosinophilic mesodermal cells in immature testes and regenerating ovaries and the morphological similarity of these cells to eosinophilic cells on the intestinal wall of males and female suggest that the mesodermal component of the new germinal zone of ovaries may result from immigration of eosinophilic cells from the surface of the intestinal wall. Indeed, we found single putatively migrating eosinophilic cells in females near the ovarian germinal zone (Fig. 3B), outside the intestinal wall (Fig. 3E) and also among adipose tissue (Fig. 3F).

Rare mesodermal cells remain in the former germinal zone of the narrow atrophied area connecting the anterior and middle sections of the ovary. Old mesodermal cells, in contrast to new mesodermal cells are flattened (Fig. S1A). We found rare nests of 3–5 eosinophilic cells on the intestinal basal membrane opposite of the atrophied ovary sections (Fig. S1B). We also found an accumulation of numerous eosinophilic cells in the anterior part of body near the intestine and on its wall (Fig. 4A) in the female with a high degree of ovarian atrophy, an event that is likely to have preceded the appearance of numerous eosinophilic cells in the female with ovarian regeneration. The same eosinophilic cells, which are in low numbers in the anterior part of the body of females with normal ovaries (Fig. 4B), may be the source of numerous eosinophilic cells in females with atrophied ovaries.

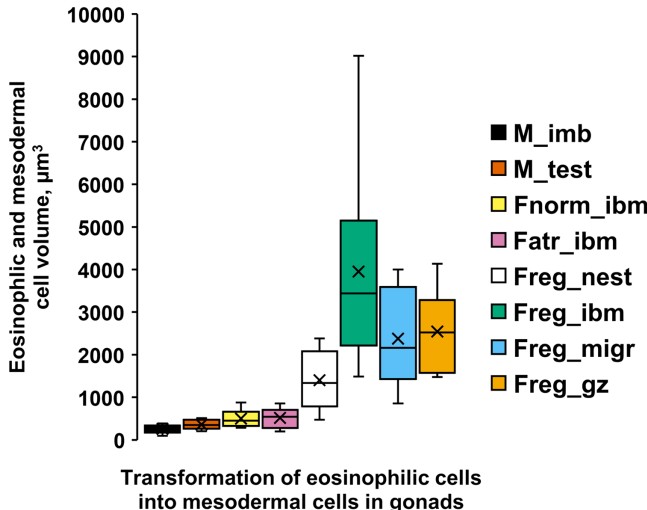

**Figure 5 Volumes (μm³) [range, upper and lower quartile (box), mean (x), median (solid line)] of eosinophilic cells (ecs) and mesodermal cells (mcs) in *A. eschrichtii* with different reproductive status of testes and ovaries.** (M_ibm): ecs on the intestinal basal membrane; and (M_test): mcs in the germinal zone of immature 13.5 mm male testes. (Fnorm_ibm): ecs on the intestinal basal membrane in the anterior part of the body in a female with normal ovaries. (Fatr_ibm): ecs on the intestinal basal membrane in the anterior part of the body in a female with ovarian atrophy. (Freg_nest): ecs in the "nests" on the intestinal basal membrane of a female with regenerating ovaries. (Freg_ibm): ecs on the intestinal basal membrane opposite the germinal zone in a female with regenerating ovaries. (Freg_migr): putatively migrating ecs outside of the intestinal basal membrane in the female with regenerating ovaries. (Freg_gz): mcs in the germinal zone of a female with regenerating ovaries. Statistical tests for differences among these cell volumes summarized in Table S1.

In the regenerating ovaries, in contrast to immature testis, the formation of the female germinal zone is accompanied by a significant increase in eosinophilic cell volumes (Fig. 5, Table S1). Ovarian atrophy in females of the second year of life, however, is not accompanied by an increase in the sizes of eosinophilic cells. Thus, the volumes of these cells in a female with normal ovaries (Fig. 5, Fnorm_ibm) and in a female with atrophied ovaries (Fig. 5, Fatr_ibm) were similar. Migrating in the posterior direction, eosinophilic cells settle on the intestinal wall of a 3-year-old female and form nests of 3–5 cells on it. The sizes of eosinophilic cells in the nests (Fig. 5, Freg_nest) increase in comparison with the previous values. The regeneration of the ovaries begins from their posterior. The epithelium, which appears on the intestinal wall opposite the posterior of the ovaries, due to the proliferation of eosinophilic cells in the nests, supplies migratory cells that populate the ovary and transform into mesodermal cells of the new germinal zone. The volumes of eosinophilic cells in the epithelium (Fig. 5, Freg_ibm), as well as cells migrating to the ovary (Fig. 5, Freg_migr.) and mesodermal cells in the new germinal zone (Fig. 5, Freg_gz) reach their maximum values for the entire period of ovarian regeneration.

## DISCUSSION

Regeneration of *A. eschrichtii* ovaries appears to result from proliferation of germ cells left over from the previous ovary and of eosinophilic cells of mesodermal origin, that migrate

into the ovary from the surface of intestinal wall. Regeneration of third-year *A. eschrichtii* ovaries appears to begin in the posterior ovary. In our observed female, the posterior region was restored and filled with new primary follicles. The middle region of the ovary included a new germinal zone while the anterior region of the ovary remained an empty connective tissue tube—a sign of previous ovarian atrophy (*Durkina, Chapman & Demchenko, 2018*). *Durkina, Chapman & Demchenko (2018)* results suggested that ovarian regeneration could occur in the third year females. The third-year female ovary reported here contained large vitellogenic oocytes and thus, given sufficient food, third-year females appear competent to regenerate ovaries and produce a second brood.

The new germinal zone and the formation of primary follicles are consistent with females having sources of germ and mesodermal cells for ovarian regeneration. Our results herein indicate that germ cells preserved in the atrophied ovaries of second-year females can produce cells of a new germinal line, as in other invertebrates and vertebrates (*Lin, 1998*; *Dansereau & Lasko, 2008*; *Dunlop, Telfer & Anderson, 2014*; *Grieve et al., 2015*; *Truman, Tilly & Woods, 2017*). Survival of germline stem cells (GSCs) was demonstrated in starvation experiments with the nematode *Caenorhabditis elegans* (*Angelo & Van Gilst, 2009*). Surviving GSCs regenerate a new germinal line when the animals are fed again. Fasting extends the reproductive longevity of these nematodes (*Angelo & Van Gilst, 2009*). Three year old *A. eschrichtii* females may thus also be products of starvation.

The source of mesodermal cells is outside the regenerating ovaries, since there is no proliferation of these cells in the ovaries. Flattened mesodermal cells remaining in the ovaries of the 3 year *A. eschrichtii* may be too rare to form new germinal zone and a new follicular epithelium. The source of mesodermal cells can be eosinophilic cells, which are initially located in small numbers in the anterior part of the body of females with normal ovaries. We found an increased accumulation of eosinophilic cells in the anterior part of the 24 mm female body (with more pronounced ovarian atrophy) (Fig. S2), but not in the 27 mm female (with less pronounced ovarian atrophy). Ovarian atrophy coincident with starvation, may thus stimulate the proliferation of these cells. The largest numbers of these cells in the female with regenerating ovaries were on the intestinal wall opposite the new germinal zone (Fig. S2). A consecutive change in the localization of the eosinophilic cells in the body of females with different states of the ovaries is consistent with anterior posterior migration along the intestine. The free-lying eosinophilic cells next to the regenerating ovaries are consistent with their migration from the surface of intestinal wall into the ovaries and with their subsequent formation of the mesodermal component of the new germinal zone. Cell migration, growth in cell size, and increases in the number of cells are common processes during tissue regeneration (*Santos et al., 2014*; *Richert et al., 2018*). Cell size can be increased by inhibiting the development of the cell cycle, increasing the rate of biosynthesis, or both (*Miettinen et al., 2014*).

Eosinophilic cells have high nuclear-cytoplasmic ratios, nuclei with condensed chromosomes but lack a nucleolus and are candidates to restore the somatic component of ovaries. High nuclear-cytoplasmic ratios and condensed chromosomes are characteristic of quiescent cells of yeasts and human fibroblasts which served as a model for the study

of the quiescence state (*Valcourt et al., 2012*; *Rumman, Dhawan & Kassem, 2015*; *Swygert et al., 2019*). Nucleoli are formed from nucleolus-organizing regions of chromosomes (*Goessens, Thiry & Lepoint, 1987*). The absence of nucleoli in the nuclei of eosinophilic cells is consistent with their disintegration during condensation of chromosomes. Quiescent cells, which include stem cells, do not undergo genome replication, have altered cellular metabolism, and are resistant to diverse stressors (*Valcourt et al., 2012*; *Rumman, Dhawan & Kassem, 2015*). Quiescence is a reversible actively maintained state in which signalling pathways can produce rapid reactivation (*Cheung & Rando, 2013*). Quiescent cells do not divide but can re-enter the cell cycle and resume proliferation if they receive the necessary microenvironmental signals (*Mohammad et al., 2019*). In particular, ovarian atrophy may be a signal for active proliferation of eosinophilic cells that become precursors of the mesodermal component in regenerating ovaries. Reactivation of quiescent cells into proliferation is crucial for tissue repair and regeneration (*Yao, 2014*).

## CONCLUSIONS

Summer starvation (*Demchenko et al., 2016*; *Durkina, Chapman & Demchenko, 2018*) may extend female *A. eschrichii* life spans and permit their reproduction in a third year. Females that form a first generation of vitellogenic eggs in summer and autumn and that breed in winter may have a sufficient reserve of nutrients to survive through the following summer. Females that breed in the summer, may spend all of their energy reserves for production of a first brood of eggs at 2 years and can use second generation of vitellogenic oocytes produced in late spring, for nutrition over the food limited summer period. Resorption of ova (atresia) leads to ovarian atrophy during extreme summer starvation. The rare females, that survive until food is again abundant, may have extended lifespans that would only be adaptive if they can restore their ovary functions.

Extended lives are seldom reported and ovarian regeneration has not been described previously in amphipods. Both adaptations may be common. These interpretations of cell origins and cell migration are from limited to data from fixed tissue photographs, and thus warrant further examination. Additional studies should include other species, more direct observations, experimental manipulations and winter samples of Sakhalin Shelf *A. eschrichtii*.

Nevertheless, *Ampelisca macrocephala* also have a predominantly 2 year life cycle with possible reproductive 3 year old females that breed a second time (*Kanneworff, 1965*). *Orchestia gammarellus* females typically live 12 to 15 months, but can survive and reproduce at approximately 36 months (*Persson, 1999*). Most females of the giant predatory amphipod, *Eusirus perdentatus*, brood only once during their approximately 6 year life span, however, rare individuals (probably reaching 8 years of age) may also carry a second brood (*Arntz et al., 1992*). *Durkina, Chapman & Demchenko (2018)* also examined a third-year female with completely regenerated ovaries and another third-year female with large vitellogenic oocytes that appeared to be competent for the upcoming reproductive season. These results thus indicate an adaptation to starvation that is consistent with *Demchenko et al. (2016)* and *Durkina, Chapman & Demchenko (2018)*

default assumption of winter feast and summer starvation. Examinations of Sakhalin Shelf *A. eschrichtii* in winter could test these hypotheses and conclusions directly. The ability of *A. eschrichtii* to regenerate the ovaries and the ability of Malacostraca, such as shrimps and crabs, to regenerate the ovaries multiple times during the life cycles, are likely due to similar mechanisms.

## ABBREVIATIONS

| | |
|---|---|
| **aof** | atresia of ovarian follicles |
| **dpvo** | degrading previtellogenic oocyte |
| **ec** | eosinophilic cell |
| **fc** | follicular cell |
| **gc** | germ cell |
| **gz** | germinal zone |
| **gzi** | islets of germinal zone |
| **ibm** | intestine basal membrane (= middle layer of intestine wall) |
| **mc** | mesodermal cell |
| **mec** | migrating eosinophilic cell |
| **ngz** | new germinal zone |
| **obm** | ovary basal membrane |
| **og** | oogonia |
| **om** | oogonial mitosis |
| **pvo** | previtellogenic oocyte |
| **t** | testis |
| **vo** | vitellogenic oocyte |

## ACKNOWLEDGEMENTS

We dedicate this paper to our mothers, Alexandra Durkina 1930–2020, Clara Jean Chapman 1926–2020, and Valentina Demchenko 1946–2020, who were our greatest mentors and advocates. We thank Cassandra G. Extavour and James Monaghan for their helpful reviews.

### Funding
The authors received no funding for this work.

### Competing Interests
The authors declare that they have no competing interests.

### Author Contributions
- Valentina B. Durkina conceived and designed the experiments, performed the experiments, analyzed the data, prepared figures and/or tables, authored or reviewed drafts of the paper, and approved the final draft.

- John W. Chapman analyzed the data, authored or reviewed drafts of the paper, and approved the final draft.
- Natalia L. Demchenko analyzed the data, prepared figures and/or tables, authored or reviewed drafts of the paper, and approved the final draft.

## Data Availability

The raw data is available in the Supplemental File.

## Supplemental Information

Supplemental information for this article can be found online at http://dx.doi.org/10.7717/peerj.12950#supplemental-information.

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
