# Peer review of "First observations of ovary regeneration in an amphipod, Ampelisca eschrichtii Krøyer, 1842"

_PeerJ, doi:10.7717/peerj.12950_

## Round 0.1 · original submission · Minor Revisions

You need to pay careful attention to all reviewers' comments and should consider these comments and corrections in submitting a revised manuscript.

·

Basic reporting

No comment - I find the manuscript meets these standards.

Experimental design

Some methods and logic underlying experimental design should be clarified, as noted in the full report.

Validity of the findings

Findings are well supported by the data provided, but their interpretation should be softened in some cases, as described in the full report.

Additional comments

Summary
This manuscript presents work inspired by the observation, made in previous studied by these authors, that female Ampelisca eschrichtii amphipods, appear able to restore ovaries to full function following starvation-induced ovarian atrophy. Such females, which would presumably normally lose reproductive ability at two years of age, had been observed to occasionally give rise to offspring at three years old. The authors ask what the source of “new” germ cells and somatic cells of the ovary might be in these females, and use classical histological techniques to address this question. They interpret their observations as suggesting that in the “regenerated” ovary, germ cells arise from previously dormant early stage oogonia, and that somatic cells arise from mesodermal cells that migrate to the ovary from their origins in the intestine.

The manuscript is clearly and concisely written, the experimental design is quite clear and sound overall, and the data of good quality and well presented. The problem addressed is extremely interesting and important for reproductive and regenerative biology in general. My only major concern is that some of the conclusions presented, namely those concerning the migration of cells, are overinterpretations of the data available, which are static snapshots of fixed tissue. It is understandable that the data are limited in this way, given the apparent rarity of this type of regeneration and the limited samples therefore available. This issue could easily be addressed by simply acknowledging in the manuscript the limitations of the data type available, and discussing alternative explanations for the observations shown.

Other minor concerns are also all easily addressable by simply adding clarifications to the manuscript, including (a) the relationship between animal length and age; (b) the evidence that animals collected from the field for observation are indeed “starving”; (c) the cytological criteria used to distinguish different cell types from one another.

Below I provide numbered explanations of these most important points; other minor suggestions for the MS are available in the marked PDF that accompanies this review.

Major Concerns
1. It is not possible to unambiguously conclude that any cells are migrating, from data that are gathered from fixed samples. Please address this in the manuscript, and soften or modify conclusions presented accordingly.
2. In a number of places in the manuscript, single examples of cellular behavior from the literature are stated as if they were universal features across animals. Examples include text in the areas of lines 71-77 and lines 218-227. Please rephrase these sections of the manuscript to make it clear in which animal(s) or cell type(s) there is evidence for these processes, in a way that accurately reflects the references cited as evidence for these statements.

Minor Concerns
1. Body length is taken as a proxy for age, but there may be limitations to this approach especially if animals are indeed starved for any period of their life. The authors should make this caveat explicit.
2. Ovary regeneration is taken to be a possible response to starvation conditions, but the evidence that the animals observed by the authors were actually starving is unclear. The authors should make this caveat explicit.
3. The criteria used to designate cells as “eosinophilic” is not clear from the Figures. For example, in Figure 1, the legend states that mesodermal cells are “intensely stained” with eosin, “in contrast” to follicular cells,” but in the micrographs shown, the cells labelled as follicular cells have darker staining than those labelled as mesodermal cells.

·

Basic reporting

The article meets standards of professional English. Just minor spelling and grammatical errors were observed. Literature references were sufficient.

Experimental design

The finding about this interesting life history story of extended reproductive success under pressure from likely limited resources the previous breeding season is interesting. A great example of how an animal can cope with environments with limited resources. I include in the comments below requests for further description of the histological methods and some brightening up of the images.

Validity of the findings

The study is observational in nature, which was done so sufficiently.

Additional comments

Below are several specific comments on errors or requests for further explanation.

Line 58 and 63: Epithelium misspelling

Line 59-61: Rework sentence structure

Line 106: Misspelling of formation

Line 123: Rework sentence structure

Methods: More information on the histological preparation is needed. Are these cryosections or paraffin sections. What was the tissue section thickness. Were all the stains hematoxylin and eosin stained and how was this performed?

Figures: The histological images could be brightened up so the background is white. This will provide more contrast for the reader, which is acceptable because it shouldn't change change any of the interpretations or conclusions made from the images.

Line 141-143: Rework sentence structure

Figure 2: How many cells were visualized?

---

## Round 0.2 · accepted · Accept

All the comments of the reviewers were taken into account and detailed comments were given to these comments. The new version of the manuscript has been corrected.

Thank you